# Fetal Cannabinoid Syndrome: Behavioral and Brain Alterations of the Offspring Exposed to Dronabinol during Gestation and Lactation

**DOI:** 10.3390/ijms25137453

**Published:** 2024-07-07

**Authors:** Daniela Navarro, Ani Gasparyan, Francisco Navarrete, Jorge Manzanares

**Affiliations:** 1Instituto de Neurociencias, Universidad Miguel Hernández-CSIC, Avda. de Ramón y Cajal s/n, San Juan de Alicante, 03550 Alicante, Spain; dnavarro@umh.es (D.N.); agasparyan@umh.es (A.G.); fnavarrete@umh.es (F.N.); 2Redes de Investigación Cooperativa Orientada a Resultados en Salud (RICORS), Red de Investigación en Atención Primaria de Adicciones (RIAPAd), Instituto de Salud Carlos III, MICINN and FEDER, 28029 Madrid, Spain; 3Instituto de Investigación Sanitaria y Biomédica de Alicante (ISABIAL), 03010 Alicante, Spain

**Keywords:** dronabinol, perinatal, fetal cannabinoid syndrome, gestation, lactation, offspring, brain, behavior, sex-dependent

## Abstract

This study establishes a fetal cannabinoid syndrome model to evaluate the effects of high doses of dronabinol (synthetic THC) during pregnancy and lactation on behavioral and brain changes in male and female progeny and their susceptibility to alcohol consumption. Female C57BL/6J mice received dronabinol (10 mg/kg/12 h, p.o.) from gestational day 5 to postnatal day 21. On the weaning day, the offspring were separated by sex, and on postnatal day 60, behavioral and neurobiological changes were analyzed. Mice exposed to dronabinol exhibited increased anxiogenic and depressive-like behaviors and cognitive impairment. These behaviors were associated with neurodevelopment-related gene and protein expression changes, establishing, for the first time, an association among behavioral changes, cognitive impairment, and neurobiological alterations. Exposure to dronabinol during pregnancy and lactation disrupted the reward system, leading to increased motivation to consume alcohol in the offspring. All these modifications exhibited sex-dependent patterns. These findings reveal the pronounced adverse effects on fetal neurodevelopment resulting from cannabis use during pregnancy and lactation and strongly suggest the need to prevent mothers who use cannabis in this period from the severe and permanent side effects on behavior and brain development that may occur in their children.

## 1. Introduction

Cannabis is the most widely used drug globally (estimated at 4% of the world’s population). In recent decades, there has been a marked increase in cannabis use among women, as reported by the World Health Organization (WHO) [1]. In particular, there has been a significant increase in daily or near-daily cannabis use during pregnancy, as revealed by the Substance Abuse and Mental Health Services Administration (SAMMA) in 2020 [2]. This spike in cannabis use can be attributed to expanding global legalization and the prevailing perception of its safety, even during pregnancy [3].

Cannabis is used during pregnancy to alleviate nausea and vomiting, while postnatally, it treats postpartum anxiety and depression [4,5]. Of particular concern is the use of cannabis during the first trimester of pregnancy, which coincides with a critical period of fetal development characterized by neurogenesis processes. The developing fetus is highly vulnerable to changes in the nervous system during this period. Notably, half of women who initiate cannabis use before pregnancy continue to use it throughout pregnancy and lactation [6].

Animal studies have shown that prenatal exposure to cannabis affects the dopaminergic, opioid, and serotoninergic systems and can disrupt the balance of excitatory and inhibitory neurotransmitter systems [7,8,9,10,11,12]. In addition, changes in the bioenergetic balance of the brain’s pyramidal neurons have been reported [13]. Tetrahydrocannabinol (THC), the psychoactive compound in Cannabis sativa, and its metabolites can cross the placenta during pregnancy due to their lipophilic nature and small molecular size [14]. Animal studies have shown that THC concentrations in fetal blood and tissues are approximately 10% lower than maternal blood levels [15]. However, in a rat model, the repeated administration of high doses of THC produced higher levels in the fetus, suggesting that the continued and excessive use of cannabis may result in significant amounts in the developing fetus [14]. Perinatal exposure to THC can lead to cognitive alterations and may induce higher morphine self-administration in offspring [16,17,18,19]. Also, THC exposure during lactation is associated with memory and social recognition deficits [20]. 

In human studies, cannabis use after the 15th week of pregnancy has been linked to a lower birth weight, a smaller head circumference, a shorter length at birth [21], an increased risk of preterm birth, and metabolic abnormalities [22,23]. Children exposed to cannabis during pregnancy showed several changes, such as increased body movements and decreased quiet sleep time [24]. They also showed decreased short-term memory and attention spans and poorer performances on memory tests, visual analysis, attention, inhibitory control, and academic performance [25]. The presence of CB1 receptors in white matter and regions where neurogenesis occurs highlights their involvement in critical neuronal developmental processes such as proliferation, migration, and synaptogenesis. THC use may alter the cannabinoid tone and its involvement in brain maturation processes. It also alters the activity of tyrosine hydroxylase, an enzyme essential for dopamine synthesis, which plays a critical role in the maturation of dopaminergic neurons [26]. However, given the exponential increase in THC concentrations in cannabis compared to two decades ago, there is a lack of longitudinal studies, which underscores the need for further research in this area.

The transfer of THC and its metabolites into breast milk also occurs during lactation [27]. Indeed, THC concentrations in breast milk in breastfeeding mothers can be up to eight times higher than those in blood plasma [28,29]. This exposure to cannabis through breast milk may be associated with impaired motor development in infants at one year of age [30]. However, the effects of cannabis during lactation on offspring remain inconclusive due to limited research in this area.

For the first time, this study utilizes an animal model to investigate the impact of dronabinol exposure during gestation and lactation on the emotional and cognitive aspects of offspring and the connection between these effects and changes in gene expression within the HPA axis, cannabinoids, and reward systems. Additionally, the research sheds light on alterations in the neurodevelopment of the cerebral cortex and hippocampus (HIPP), revealing observations related to cortical lamination, neuronal plasticity, and synaptic and inhibitory excitatory circuitry.

Moreover, the study uncovers that dronabinol exposure leads to increased motivation and alcohol consumption. Notably, all these effects manifest in a sex-dependent manner. The analysis considers the elevated THC concentrations present in contemporary cannabis, along with the frequency and chronic nature of its usage.

These findings underscore the significant adverse impact of cannabis use during pregnancy and lactation on fetal neurodevelopment. They strongly advocate for preventive measures to dissuade mothers from using cannabis during these critical periods. Such precautions are crucial to avoiding the severe side effects on behavior and brain development observed in the offspring.

## 2. Results

### 2.1. Behavioral Evaluation

#### 2.1.1. Emotional Evaluation

Male and female progeny exposed to synthetic THC during their perinatal period spent less time in the open arms of the elevated plus maze (EPM) apparatus than vehicle-exposed rodents (Males, Figure 1A: Student’s *t*-test, *p* = 0.003, *t* = 3.224, 27 d.f.; Females, Figure 1C: Student’s *t*-test, *p* = 0.001, *t* = 3.846, 20 d.f.), without affecting the number of transitions between open and closed arms (Males, Figure 1B: Student’s *t*-test, *p* = 0.05, *t* = 2.055, 26 d.f.; Females, Figure 1D: Student’s *t*-test, *p* = 0.279, *t* = 1.125, 14 d.f.). 

In the novelty-suppressed feeding test (NSFT), both males and females perinatally exposed to the THC showed significantly higher latency times (Males, Figure 1E: Student’s *t*-test, *p* = 0.009, *t* = −2.873, 23 d.f.; Females, Figure 1G: Student’s *t*-test, *p* < 0.001, *t* = −4.518, 21 d.f.) and significantly reduced food consumption during the 5-min evaluation period (Males, Figure 1F: Student’s *t*-test, *p* = 0.002, *t* = 3.527, 27 d.f.; Females, Figure 1H: Student’s *t*-test, *p* = 0.019, *t* = 2.553, 21 d.f.) compared to the VEH group. 

The tail suspension test (TST) analysis revealed an increased immobility time in males (Figure 1I: Student’s *t*-test, *p* < 0.001, *t* = −4.311, 24 d.f.) and females (Figure 1J: Student’s *t*-test, *p* < 0.001, *t* = −4.527, 18 d.f.) exposed to THC during their perinatal period compared to vehicle-exposed mice. 

#### 2.1.2. Cognitive Evaluation

In the novel object recognition test (NOR), THC-exposed male (Figure 2A: Student’s *t*-test, *p* = 0.021, *t* = 2.450, 27 d.f.) and female (Figure 2B: Student’s *t*-test, *p* = 0.006, *t* = 3.118, 17 d.f.) mice presented a lower discrimination index than controls.

The step-down inhibitory avoidance (SDIA) test showed a statistically significant reduction in the latency time of male and female THC offspring at 1 h and 24 h of evaluation (Figure 2C: One-way RM ANOVA, F (5,98) = 39.331, *p* < 0.001; Figure 2D: One-way RM ANOVA, F (5,95) = 57.432, *p* < 0.001). 

Prepulse inhibition (PPI) data were analyzed by two-way ANOVA followed by a Student–Newman–Keuls posthoc test, which revealed a significantly reduced prepulse inhibition in both males (Figure 3A: Two-way ANOVA, Treatment F(1,62) = 9.294, *p* = 0.003, Prepulse F(2,62) = 32.519, *p* < 0.001, Treatment × Prepulse F(2,62) = 1.204, *p* = 0.307) and females (Figure 3B: Two-way ANOVA, Treatment F(1,80) = 11.827, *p* < 0.001, Prepulse F(2,80) = 35.552, *p* < 0.001, Treatment × Prepulse F(2,80) = 0.932, *p* = 0.398) at 74 and 82 dB, but not at 90 dB, in comparison with VEH mice. 

### 2.2. Gene Expression Studies under Basal Conditions

#### 2.2.1. Stress Axis

Under basal conditions, *Crf* gene expression in the PVN was significantly greater in females than in males (Figure 4A: Student’s *t*-test, *p* = 0.037, *t* = −2.213, 23 d.f.). Interestingly, the perinatal THC exposure did not modify the *Crf* expression in males (Figure 4B: Student’s *t*-test, *p* = 0.144, *t* = 1.515, 22 d.f.) but induced a downregulation in females (Figure 4C: Student’s *t*-test, *p* < 0.001, *t* = 3.789, 24 d.f.). The gene expression analysis of *Nr3c1* in the HIPP revealed higher basal levels in females than in males (Figure 4D: Student’ *t*-test, *p* = 0.002, *t* = −3.470, 22 d.f.). Exposure to dronabinol during gestation and lactation increased *Nr3c1* gene expression in males (Figure 4E: Student’s *t*-test, *p* < 0.001, *t* = −3.851, 20 d.f.) but not in females (Figure 4F: Student’s *t*-test, *p* = 0.547, *t* = 0.611, 24 d.f.). 

#### 2.2.2. Reward Circuit

In the NAcc of females, the expression of the *Oprm1* gene was decreased compared to that in males (Figure 4G: Student’s *t*-test, *p* = 0.04, *t* = 2.186, 22 d.f.). Males (Figure 4H: Student’s *t*-test, *p* = 0.017, *t* = 2.584, 21 d.f.) and females (Figure 4I: Student’s *t*-test, *p* = 0.006, *t* = 3.076, 20 d.f.) showed a reduction in *Oprm1* gene expression after the perinatal THC exposure compared with vehicle-exposed mice. Under baseline conditions, the expression of the *Th* gene in the VTA was also reduced in females compared to males (Figure 4J: Student’s *t*-test, *p* = 0.027, *t* = 2.364, 22 d.f.). A sex-dependent differential regulation was found in mice perinatally exposed to THC. A reduction was found in males (Figure 4K: Student’s *t*-test, *p* < 0.001, *t* = 4.201, 20 d.f.) and an increase was found in females (Figure 4L: Student’s *t*-test, *p* = 0.019, *t* = −2.526, 24 d.f.). 

#### 2.2.3. Endocannabinoid Targets

There was a significantly reduced *Cnr1* gene expression in females compared to that in males in HIPP (Figure 5A: Student’s *t*-test, *p* = 0.029, *t* = 2.322, 22 d.f.). The perinatal THC exposure induces differential regulation in the gene expression of *Cnr1* in males and females, with a reduction in males (Figure 5B: Student’s *t*-test, *p* = 0.035, *t* = 2.260, 21 d.f.) and an upregulation in females (Figure 5C: Student’s *t*-test, *p* = 0.002, *t* = −3.488, 23 d.f.). Interestingly, in the AMY, no significant differences were found under baseline conditions between males and females (Figure 5D: Student’s *t*-test, *p* = 0.169, *t* = 1.421, 22 d.f.). However, THC perinatal exposure increased *Cnr1* gene expression in males (Figure 5E: Student’s *t*-test, *p* = 0.038, *t* = −2.212, 22 d.f.) and females (Figure 5F: Student’s *t*-test, *p* < 0.001, *t* = −4.039, 22 d.f.). 

In HIPP and AMY, females showed higher *Cnr2* gene expression under basal conditions (HIPP, Figure 5G: Student’s *t*-test, *p* = 0.02, *t* = −2.527, 20 d.f.; AMY, Figure 5J: Student’s *t*-test, *p* = 0.004, *t* = −3.343, 17 d.f.). Dronabinol exposure during the perinatal period increased *Cnr2* gene expression in the HIPP and AMY in males (HIPP, Figure 5H: Student’s *t*-test, *p* = 0.024, *t* = −2.475, 17 d.f.; AMY, Figure 5K: Student’s *t*-test, *p* = 0.027, *t* = −2.376, 21 d.f.). However, in females, a significant increase in the *Cnr2* gene expression was only observed in the HIPP (Figure 5I: Student’s *t*-test, *p* = 0.014, *t* = −2.721, 19 d.f.) but not in the AMY (Figure 5L: Student’s *t*-test, *p* = 0.265, *t* = −1.146, 20 d.f.). 

### 2.3. Immunohistochemistry

#### 2.3.1. NeuN Immunolabeling in the HIPP’s DG and CA in the Somatosensorial Cortex

The low magnification of the NeuN-immunostaining section (Figure 6 and Figure 7) showed an abnormal laminar organization of the HIPP in THC pups. In the THC group, the boundaries of the CA_1_ pyramidal and granular layers of the dentate gyrus (DG) with the adjacent oriens and proximal molecular layers, respectively, were more blurred than in VEH pups.

In both THC males and females, a decrease in the pyramidal layer of the CA_1_ and CA_3_ (arrow in Figure 6 and Figure 7) and the granular layer and hilus of the dentate gyrus (DG) was observed by NeuN-ir. Indeed, the somatosensory cortex of THC males and females was smaller than that of the control pups (Figure 6 and Figure 7). The cortical lamination revealed a higher density of neurons in layer VI and a lower density in layers II–III in THC pups according to the migration pattern of the neocortex. The boundaries between cortical layers are not well defined in male and female THC pups. The normal brain maturation and development of normal brain functions are affected.

#### 2.3.2. Immunolabeling BDNF, NF200, VGluT1, and VGAT in the DG and CA of the HIPP

A lower intensity of labeling the cells in all areas of the HIPP in the THC pups compared to that in the VEH male mice has been observed at low magnification by brain delivery neurotrophic factor (BDNF) immunostaining (Figure 8). In contrast, a reduction in the number and labeling intensity of BDNF-ir cells was found in the hilus and granular layer of DG areas (arrowhead in Figure 8) and the pyramidal layer of CA_1_ and CA_3_ in THC pups compared to that in VEH (Figure 8 A,B). The BDNF-ir pyramidal cells in CA3 were reduced in THC compared to those in VEH male mice (arrow in Figure 8). In female mice, the number and intensity of BDNF-ir cells in the CA_3_ decrease in THC pups compared to those in VEH mice (Figure 9). 

The labeling intensity decreased in THC males and females compared to that in VEH mice in the granular layer and hilus of the DG in the HIPP (Figure 8 and Figure 9 observed at high magnification). All these results revealed a reduction in plasticity and cell survival in THC male and female pups compared with those in VEH mice. 

NF200-ir decreased in all areas of the HIPP in THC mice compared to that in VEH male and female offspring (Figure 10 and Figure 11). Remarkably, the decrease in the NF200-ir of CA_1_ stratum radiatum and lacunosum-moleculare (arrow and double arrow in Figure 10B) and the hilus of DG (arrowhead in Figure 10B) was more evident in the THC group than in the VEH male pups. In female mice, the NF200-ir decreased in the lacunosum-moleculare of CA_1_ (arrow in Figure 11B) and the hilus of DG (arrowhead in Figure 11B) in THC pups compared with that in VEH mice. Hilus’s high magnification in the DG shows a decrease in the NF200-ir in the THC mice compared to that in the VEH male mice (Figure 10D). The hilus of the DG showed reduced NF200-ir in THC mice compared to that in VEH female mice (Figure 11D). These results suggest an alteration in male and female THC mice; the stabilization and maturation of connections in the THC mice were more pronounced compared to those in the VEH group.

Low-power confocal micrographs obtained by superimposing three consecutive optical sections (covering a depth of 5 μm) indicated that the distribution of both VGluT1-ir and VGAT-ir in the DG, CA_3_, and CA_1_ of male and female THC mice was aberrant, revealing an irregular laminar distribution of excitatory and inhibitory inputs (Figure 12 and Figure 13). The most prominent findings were a pronounced decrease in the density of VGluT1-ir in the DG distal-inner molecular layer and lacunosum-moleculare of the CA_1_ in THC males (arrowheads and arrow Figure 12A,B,D,E) and a smaller lucidum of CA_3_ in the THC male group than in the VEH pups (double arrowhead in Figure 12 A,B,D,E). In THC male mice, a reduction in the VGAT-ir was found in the lacunosum-moleculare of CA_1_ compared to that in the VEH group (arrow in Figure 12C,F). At a high amplification, a decrease in the area of VGluT1-ir mossy boutons (arrow in Figure 14) and the pyramidal layer (double arrow in Figure 14C,F) of CA_3_ was found in the THC group compared to the VEH mice. The VGluT1-ir bouton in THC female mice decreased in the strata oriens, lucidum, and radiatum of CA_3_ (double arrowhead in Figure 13) and in the DG distal-inner molecular layer (arrowhead in Figure 13A,B,D,E) stratum lacunosum-moleculare of CA_1_ in comparison to that in VEH female mice (arrow in Figure 13A,B,D,E). In particular, in the lacunosum-moleculare of CA1 (arrow in Figure 13C), VGAT-ir was decreased compared to that of the VEH group. At a high magnification, the area of VGluT1-ir mossy boutons in the stratum oriens and lucidum of CA3 was more reduced in the THC group than in the VEH female mice (arrow in Figure 15B,E), and VGAT-ir decreased in the pyramidal layer of CA3 (double arrow in Figure 15C,F).

### 2.4. Oral Ethanol Self-Administration (OEA)

The statistical analysis revealed that male mice exposed to THC during gstation and lactation showed an increased number of effective responses (Figure 16A: Two-way RM ANOVA, FR1, Treatment F(1,144) = 9.931, *p* = 0.004, Day F(4,144) = 6.079, *p* < 0.001, Treatment x Day F(4,144) = 0.402, *p* = 0.807; FR3, Treatment F(1,144) = 4.628, *p* = 0.041, Day F(4,144) = 1.641, *p* = 0.169, Treatment × Day F(4,144) = 0.0921, *p* = 0.985) and ethanol intake (Figure 16B: Two-way RM ANOVA, FR1, Treatment F(1,144) = 9.542, *p* = 0.005, Day F(4,144) = 4.632, *p* = 0.002, Treatment × Day F(4,144) = 0.545, *p* = 0.703; FR3, Treatment F(1,144) = 6.998, *p* = 0.013, Day F(4,144) = 0.709, *p* = 0.587, Treatment × Day F(4,144) = 0.800, *p* = 0.528) in FR1 and FR3, compared with those in VEH males. Further analysis showed an increased breaking point (Figure 16C: Student’s *t*-test, *p* = 0.039, *t* = −2.212, 19 d.f.) and ethanol intake (Figure 16D: Student’s *t*-test, *p* = 0.006, *t* = −3.081, 19 d.f.) in the PR stage of the OEA paradigm in THC-exposed males in comparison with those of vehicle-exposed mice. 

THC female offspring had an increased number of effective responses (Figure 17A: Two-way RM ANOVA, FR1, Treatment F(1,104) = 11.817, *p* = 0.003, Day F(4,104) = 12.493, *p* < 0.001, Treatment × Day F(4.053, *p* = 0.005; FR3, Treatment F(1,104) = 6.512, *p* = 0.019, Day F(4,104) = 1.599, *p* = 0.183, Treatment × Day F(4,104) = 0.0562, *p* = 0.994) and ethanol intake (Figure 17B: Two-way RM ANOVA, FR1, Treatment F(1,104) = 87.981, *p* < 0.001, Day F(4.104) = 3.414, *p* = 0.013, Treatment × Day F(4,104) = 0.980, *p* = 0.424; FR3, Treatment F(1,104) = 6.264, *p* = 0.022, Day F(4,104) = 1.654, *p* = 0.170, Treatment × Day F(4,104) = 0.363, *p* = 0.834) in the FR1 and FR3 phases of the OEA paradigm compared to those of nonexposed females. Interestingly, no differences were found in the breakpoint (Figure 17C: Student’s *t*-test, *p* = 0.598, *t*= −0.536, 19 d.f.) nor in the ethanol intake (Figure 17D: Student’s *t*-test, *p* = 0.805, *t* = −0.250, 19 d.f.) of THC-exposed females compared to those of VEH mice.

#### Gene Expression Studies in the OEA Paradigm-Exposed Mice 

In the NAcc and after the OEA paradigm exposure, females showed an increase in the relative gene expression of *Cnr1* in comparison with males (Figure 18A: Student’s *t*-test, *p* = 0.016, *t* = −2.618, 20 d.f.). Interestingly, males exposed to THC during gestation and lactation showed higher *Cnr1* gene expression (Figure 18B: Student’s *t*-test, *p* = 0.016, *t* = −2.599, 22 d.f.), whereas it was reduced in females (Figure 18C: Student’s *t*-test, *p* = 0.009, *t* = 2.903, 20 d.f.). *Cnr2* receptor gene expression was increased in females exposed to the OEA paradigm compared with that in males (Figure 18D: Student’s *t*-test, *p* = 0.007, *t* = −3.052, 18 d.f.). Interestingly, for males (Figure 18E: Student’s *t*-test, *p* < 0.001, *t* = −6.288, 22 d.f.) and females (Figure 18F: Student’s *t*-test, *p* < 0.001, *t* = −5.733, 18 d.f.), perinatal THC exposure also increased *Cnr2* gene expression in the NAcc.

The OEA paradigm exposure decreased *Oprm1* gene expression in females under baseline conditions compared to males (Figure 18G: Student’s *t*-test, *p* < 0.001, *t* = 6.163, 20 d.f.). The perinatal exposure to the THC did not induce any modification in males after the OEA paradigm (Figure 18H: Student’s *t*-test, *p* = 0.119, *t* = 1.622, 22 d.f.). However, an increase in *Oprm1* gene expression was found in females perinatally exposed to the THC compared with vehicle-exposed females (Figure 18I: Student’s *t*-test, *p* = 0.002, *t* = −3.473, 20 d.f.). 

Under basal conditions in the VTA, a significant statistical reduction in relative Th gene expression was observed in females exposed to the OEA paradigm compared to males (Figure 18J: Student’s *t*-test, *p* = 0.004, *t* = 3.280, 20 d.f.). The perinatal exposure to the THC and the OEA paradigm induced the sex-dependent regulation of this target’s gene expression, with an increase in males (Figure 18K: Student’s *t*-test, *p* = 0.001, *t* = −3.735, 22 d.f.). 

## 3. Discussion

The results of this study show that dronabinol consumption during gestation and lactation induces pronounced behavioral and neurobiological changes in the offspring. The facts in support of this statement are as follows: (1) male and female mice exposed to dronabinol during gestation and lactation present significant emotional hyperreactivity with anxiogenic and depressive-like behavior; (2) mice developed cognitive disturbances (problems in associative learning memory and attentional deficit); (3) these behavioral alterations were associated with gene expression changes in several brain targets (*Crf*, *Nr3c1*, *Opmr1, Th*, *Cnr1,* and *Cnr2*); (4) males and females exposed to dronabinol during gestation and lactation displayed changes in the number of cells, neuroplasticity, neurofilaments, GABAergic, and glutamatergic neurotransmission in the HIPP and cortical lamination; (5) motivation and ethanol consumption are increased in males and females exposed to dronabinol, and this behavior has been associated with gene expression changes in the endocannabinoid (*Cnr1* and *Cnr2*) and reward (*Opmr1* and *Th*) systems.

Several animal models of THC consumption during gestation or lactation have been reported to determine potential alterations in offspring [11,12,18,20,31,32,33,34,35,36]. However, this is the first model where the elevated THC concentration found in current cannabis and the frequency and chronic nature of its use have been considered. For instance, in the 1990s, the concentration of THC was 2–6%, and today, it often reaches between 17 and 28% [37]. Unfortunately, cannabis is currently the most widely used drug of the 21st century, and due to the growing demand for recreational activities, consumption trends are rapidly increasing [38].

In this study, the dronabinol administration was 10 mg/kg twice a day, in contrast to previous reports where the THC dose was much lower and administered only once a day. In most of the studies, the administration of THC was given only during pregnancy or during pregnancy and the first lactation period, not considering both periods together [12,18,33,35,39,40,41,42,43,44]. Therefore, this model examines the effects of dronabinol (synthetic THC), the main psychoactive component of cannabis, during pregnancy and lactation, taking into account the high concentrations of THC found in cannabis today. The research goes beyond behavioral changes to examine genetic and protein changes in several brain regions, including the PVN, VTA, NAcc, HIPP, AMY, and parietal cortex. Gene expression analyses by real-time PCR were performed to assess THC consumption-induced changes in specific key targets implicated in cannabinoid addiction, such as the Nacc and VTA, for their involvement in the brain reward system [45,46]. In addition, the endocannabinoid system may regulate hormonal and behavioral responses to stress involving the *Crf* gene expression in the PVN neurons and their receptors [47]. Because of the emotional and cognitive changes found in the offspring, the expression of CB1 and CB2 receptors in the PVN and amygdala was also evaluated. Additionally, we have explored the vulnerability to ethanol self-administration of males and females from mothers receiving dronabinol during gestation and lactation.

In mice exposed to THC during the perinatal period, little information is available on changes in the HPA axis. Rubio et al. reported that females exposed to THC during pregnancy and lactation had higher levels of *Crf*. In contrast, males exposed to THC had lower levels in the medial basal hypothalamus (MBH) [32]. However, it is known that the administration of low to moderate doses of THC (5 mg/kg p.o. or 4 mg/kg s.c.) during pregnancy and part of lactation is known to induce anxiogenic behavior in male rat pups [35,42,44,48]. 

Gene and protein expression studies investigated the neurobiological mechanisms underlying dronabinol-induced behavioral changes in offspring. In this study, significant anxiety- and depressive-like behaviors were found in male and female pups exposed pre- and postnatally to dronabinol. The total number of entries in the EPM test was unaffected, indicating that perinatal dronabinol treatment did not alter locomotor activity in the offspring. The relative gene expression of *Crf* in the PVN and the glucocorticoid receptor (*Nr3c1*) in the HIPP, the first and last steps of the HPA axis, respectively, were evaluated to detect changes that may help to elucidate, at least in part, the emotional alterations caused by the dronabinol model. Under basal conditions, *Crf* gene expression was more significant in females than in males, which aligns with a previous study [49], whereas *Nr3c1* gene expression was significantly higher in the HIPP of females than in males. However, in animals exposed perinatally to dronabinol, *Crf* gene expression was significantly decreased in females, and *Nr3c1* gene expression was significantly increased in males but not altered in females. These findings of HPA axis targets suggest the impaired sex-dependent negative feedback regulation of this stress circuit since *Crf* gene regulation is affected in females and the *Nr3c1* gene is affected in males. This differential altered regulation in the HPA feedback mechanisms induced by dronabinol may be related to a distinct action of androgens and estrogens circulating hormones during the development of this stress circuit. Gonadal hormones play a crucial role in regulating the HPA axis. They influence the response and sensitivity to releasing factors, neurotransmitters, and hormones. From early in life, gonadal steroids can have differential effects on the HPA axis, resulting in sex differences in the responsiveness of this axis. [50]. Nevertheless, further studies are needed to determine the factors that may cause these differences. 

Cannabinoid receptors have been implicated in the regulation of anxiety, stress, and cognition [51]. In this study, mice exposed to dronabinol during pregnancy and lactation showed sex differences in regulating cannabinoid receptor gene expression. This was reflected by decreased expression in males and increased expression in females in HIPP. However, in AMY, *Cnr1* expression improved in both males and females. 

Under basal conditions, *Cnr2* gene expression was markedly higher in the HIPP and AMY of females than in males. Dronabinol exposure during gestation and lactation increased *Cnr2* gene expression in the HIPP of males and females but only in the AMY of males. Alterations in *Cnr2* gene expression might be related to emotional dysregulation, with anxiogenic and depressive behaviors and emotional hyperreactivity, as found in the behavioral evaluations. 

The dysregulation of HPA axis activity may be associated with altered cannabinoid receptors in male and female mice. The sexual differences found in cannabinoid receptors function under basal conditions and in the HIPP and AMY of mice exposed to dronabinol may be related, at least in part, to the influence of circulating sexual hormones and their distinct regulation of these two brain regions. 

Attentional deficit and cognitive impairment have been reported after the chronic consumption of THC [52,53]. Prepulse inhibition test (PPI) results showed that male and female mice exposed to dronabinol during pregnancy and lactation exhibited a pre-attentional deficit. Notably, a decrease of 40% in the startle amplitude was found in males and females of THC pups compared with the control group. Immunohistochemical analyses focused on studying the neocortex and hippocampal architecture to understand, at least in part, the response to the PPI test. The results of NeuN-ir revealed that the somatosensory cortex is smaller in dronabinol mice than in control pups. Indeed, there is a higher density of neurons in layer VI and a lower density in layers II-III in dronabinol pups according to the migration pattern of the neocortex. The boundaries between cortical layers are poorly defined in male and female dronabinol pups. The irregularities in cell migration and changes in cytoarchitectural organization found in dronabinol offspring suggest that the normal process of brain maturation, cortical lamination, and establishing normal brain functions will likely be altered. Therefore, it is reasonable to hypothesize that the continued administration of dronabinol during prenatal stages may lead to alterations in the normal developmental patterns of offspring. Morphophysiological abnormalities established during gestation and early postnatal life may be preserved throughout life and may be a risk factor for the development of neuropsychiatric disorders later in life.

Defects in the PPI of the startle reflex are among the most commonly used physiological markers in animal models of schizophrenia or psychosis [54]. The mammalian acoustic startle response is generated by a relatively simple neural circuit in the lower brainstem [55], where HIPP is also involved [55,56,57,58]. The results of this study also showed a decrease in the pyramidal layer of the CA_1_ and CA_3_ (arrow in Figure 6 and Figure 7), and the granular layer and hilus of the dentate gyrus (DG) were observed by NeuN-ir both in male and female mice exposed to dronabinol during pregnancy and lactation compared to controls. In both sexes, the offspring showed irregular laminar organization of the HIPP, which may be at least partially related to the attentional deficit.

Significant sex-dependent cognitive changes were also found in the offspring exposed to dronabinol from GD5 to PND21. Male mice performed worse than females on the NOR discrimination index, although both showed impaired spatial memory. Males and females showed impaired aversive memory in both the short and long term. However, males display more pronounced memory impairment than females. These cognitive alterations may be related to synaptogenesis and neuroplastic changes as well as GABAergic and glutamatergic transmission alterations found by immunohistochemical analyses performed in the HIPP. The BDNF-TrkB pathway is central to activity-dependent synaptic plasticity associated with learning and memory [59]. Impaired synaptic plasticity has been reported at glutamatergic synapses in diseases where BDNF function is compromised [60]. Accordingly, immunohistochemistry for NF200 revealed an altered stabilization and maturation of both neuronal connections in the exposed male and female groups compared to controls.

The DG receives its primary input from layer II of the entorhinal cortex via the perforant pathway in the HIPP. DG granule cells project to the proximal apical dendrites (stratum lucidum) of CA_3_ pyramidal cells, which in turn project through the Schaffer collaterals to the ipsilateral apical dendrites (stratum radiatum) of CA_1_ pyramidal cells. In addition, an associative excitatory network connects CA_1_ to ipsilateral CA_3_ and DG through recurrent connections and to contralateral CA_3_ and CA_1_ through commissural connections [61]. The immunohistochemistry for VGluT1/VGAT revealed a decrease in VGluT1-ir density in the DG distal-inner molecular layer and lacunosum-moleculare of CA_1_ in both male and female dronabinol pups. There was a decrease in VGluT1-ir mossy boutons in the pyramidal layer of CA3 in dronabinol pups compared to VEH mice. The findings indicate that excitatory synaptic circuitry affects male and female offspring exposed to dronabinol during neurodevelopment.

VGAT-ir boutons were decreased in the lacunosum-moleculare of CA_1_ and the pyramidal layer of CA_3_ of male and female mice after exposure to dronabinol during gestation and lactation compared to the VEH group. This change in the dronabinol pups could result in a decrease in perisomatic inhibition and an asynchronized control of the firing of the neurons. Cortical asynchrony hyperactivity and HIPP may be related to the pathophysiological basis of attention deficit hyperactivity disorder (ADHD) and are associated with deficits in working memory processing and impulsive symptoms [62]. Interestingly, as explained above, the dronabinol group shows alterations in associative memory and attention deficit. Cognitive impairment in males and females was related to neuron loss, reduced synaptic plasticity, and changes in the HIPP’s excitatory and inhibitory synaptic circuitry. All data are according to previous work that found deficits in short- and long-term memory, attention, and recognition of memory with THC administration during pregnancy [41,43,63,64]. For the first time, we reported morphological consequences in the HIPP of dronabinol exposure during pregnancy and lactation in the offspring. 

The continuous administration of dronabinol during pregnancy and lactation could indirectly affect the reward system of the offspring [12,33]. To observe how this system is affected, we measured the gene expression of *Opmr1* in the NAcc and the enzyme involved in dopamine synthesis *Th* in the VTA. Our study observed a sexual regulation of both genes with possible consequences for reward mechanisms, as observed in previous studies [12,64,65]. Under basal conditions, *Opmr1* and *Th* gene expression was significantly reduced in females compared to males in the NAcc and VTA, respectively. Specifically, real-time PCR gene expression studies showed lower *Opmr1* gene expression in the NAcc of both males and females in the dronabinol treatment group compared to the VEH group. However, *Th* gene expression decreased in males and increased in females in the VTA group compared to that in the VEH group. Therefore, dronabinol exposure during pregnancy and lactation may result in functional changes in the mesolimbic dopaminergic system. This effect may underlie the increased ethanol consumption and motivation demonstrated in the OEA paradigm. The increased ethanol consumption and motivation in the OEA paradigm may be due to this effect.

The OEA paradigm was used to assess the effects of dronabinol administration during pregnancy and lactation on motivation and alcohol consumption in offspring. Dronabinol treatment increased the number of effective responses and ethanol intake in the FR1 and FR3 stages of the OEA in both male and female offspring. However, during the PR phase (the maximum expression of motivation), the breakpoint values and ethanol intake increased only in males and not in females exposed to dronabinol. These results suggest that estrogens may play an essential role in regulating the motivation for ethanol in female mice [66]. 

To investigate the neurochemical mechanisms underlying the increased susceptibility to the rewarding effects of ethanol in mice exposed to dronabinol during pregnancy and lactation, gene expression analyses were performed for *Cnr1*, *Cnr2*, *Oprm1*, and *Th* in the NAcc and VTA, respectively. The reinforcing and rewarding properties of ethanol are mediated, at least in part, by dopaminergic pathways originating in the VTA and primarily projecting to the NAcc [67,68,69], both of which undergo functional changes after chronic ethanol administration [70]. Under basal conditions, *Th* and *Oprm1* gene expressions were significantly lower in females than in males in the VTA and NAcc. In male and female mice subjected to the OAS paradigm, there was a sex-dependent variation in Th expression, significantly increased in males and decreased in females compared to VEH in VTA [70,71]. 

However, ethanol intake also facilitates the release of opioid peptides, modulating dopamine transmission within the mesocorticolimbic system [72]. Therefore, an ethanol-induced reward involves the opioidergic system [73]. In our study, OEA significantly upregulated the *Opmr1* gene expression in the NAcc of female mice but was without effect in males. This opposite finding may be partly due to differences in hormonal regulation between males and females and the differential alteration in the dopaminergic system. Further studies are required to confirm this hypothesis. 

CB1r and CB2r play a crucial role in regulating ethanol intake, tolerance, dependence, withdrawal, and relapse and in modulating the reinforcing and motivational effects of ethanol. In this study, a significant increase in the relative gene expression of *Cnr1* and *Cnr2* was observed in female mice compared to male mice under basal conditions. A sex-dependent regulation in the expression of both genes was found since dronabinol male mice presented a significant increase in *Cnr1* and *Cnr2* in the NAcc. However, in females, there was only an increase in *Cnr2* expression and a decrease in *Cnr1*. Thus, these results suggest an increased susceptibility to the rewarding properties of ethanol in the offspring of mothers consuming dronabinol during pregnancy and lactation, which depends, at least in part, on the circulating hormonal regulation of *Cnr1* and *Cnr2* gene expression in the NAcc.

The present study provides crucial information on the effects of dronabinol use on offspring during gestation and lactation. This research establishes an association among behavioral changes, cognitive impairment, and alterations in the expression of genes associated with the HPA axis, the cannabinoid system, and reward systems. Furthermore, the study underscores significant modifications occurring in the neocortex and HIPP during neurodevelopment. 

Several studies in animal models, both in rats and mice, have shown that the effects of cannabinoid use during pregnancy and lactation on offspring are long-lasting. Indeed, specific deficits in social interaction in adolescence, adulthood, and older ages were found (see review [64]). Further studies are needed to evaluate the long-term changes produced by administering dronabinol during gestation and lactation and the neurochemical mechanisms involved.

Offspring exposed to dronabinol show alterations in the HPA axis, leading to behavioral alterations. Additionally, exposure to dronabinol during pregnancy and lactation disrupts the reward system, potentially serving as a catalyst for heightened susceptibility and motivation for alcohol consumption in the offspring, and surprisingly, these effects are manifested in a sex-dependent manner. 

Conversely, the HIPP structure and cortical lamination were altered, accompanied by reduced synaptic plasticity and cell survival, observed by a BDNF immunostaining that showed a lower labeling intensity of the cells in all areas of the HIPP in THC pups compared with that in VEH male and female mice. There is also an imbalance in excitatory and inhibitory circuits. 

In human studies, there are three central longitudinal studies, such as the Ottawa Pregnancy Prospective Study (OPPS), which began in 1978, the Maternal Health Practices and Child Development (MHPCD), which started in 1982, and the Generation R study, which began in 2001, which examined the neurobehavioral consequences of maternal cannabis use. The OPPS and MHPCD studies reported poor performance in verbal and memory function [74,75,76,77], but the Generation R study did not find this impairment. In addition, 18–22-year-old offspring who underwent prenatal cannabis exposure had altered neural function in several brain areas during both visuospatial working memory processing and response inhibition [78]. However, there are some discrepancies in the results, which may be due to differences in THC doses, heavy cannabis use, and the small sample sizes of each longitudinal study, although there is an association between this drug and changes in frontal lobe function [79]. 

In conclusion, this animal model, considering the current level of consumption and the increased concentration of THC found in the plant, highlights the potential risks that may affect fetal neurodevelopment associated with cannabis use during pregnancy and lactation. However, further animal and clinical studies are needed to confirm these risks. Preventive measures are necessary to discourage mothers from using cannabis during these critical periods and avoid the potential occurrence of fetal cannabinoid syndrome.

## 4. Materials and Methods

### 4.1. Animals

We used 121 female and 32 male C57BL/6J mice of 5 weeks of age purchased from Charles River laboratories (Lille, France). After their arrival, mice were individualized and left undisturbed for one week to acclimate to the animal housing room. A total of 260 offspring (134 males and 126 females) were obtained from the crossbreeding. All animals were maintained in the Animal Facilities Service of the Universidad Miguel Hernandez (SEA-UMH) under controlled environmental conditions in terms of temperature (23 ± 2 °C), humidity (60 ± 10%), and a 12 h light–dark cycle (lights on from 08:00 to 20:00 h). All experimental procedures complied with the Spanish Royal Decree 53/2013, the Spanish Law 32/2007, and the European Union Directive of 22 September 2010 (2010/63/UE) regulating the care of experimental animals and were approved by the Ethics Committee of Miguel Hernandez University. The animal studies are reported in compliance with the ARRIVE guidelines [80,81].

### 4.2. Perinatal Dronabinol Exposure Procedure

Vaginal smears were collected in the afternoon for cytological evaluation for mouse estrous cycle staging identification. Those female mice at the pro-estrus or estrus stage of the cycle were placed into the cage of a singly housed male overnight, and the presence of a vaginal plug was monitored early in the morning. Pregnancy was determined by monitoring weight gain every 2 days and conducting vaginal cytological evaluation. From GD5 until PND21, the mothers were administered 10 mg/kg of dronabinol (THC Pharm, Frankfurt, Germany) dissolved in sunflower oil (p.o., twice a day) or the corresponding vehicle (VEH; sunflower oil, p.o., twice a day) (Figure 19). However, once the female was pregnant, the continued administration of THC resulted in pregnancy loss. Although some females completed gestation, sometimes maternal care was insufficient for offspring survival. In the group of females exposed to dronabinol, only 60% adequately completed the gestation and lactation periods. In contrast, in the control group, 100% of the initial females did so. For THC offspring that survived and were maintained throughout gestation and lactation, the litter size and weight were monitored and did not differ from those of the VEH offspring.

One week (PND 28) after the pups were weaned (PND 21), the behavioral testing of the males and females began. The tests lasted two to three days, depending on the number of animals, and were performed every 2 or 3 days. Six different behavioral tests were performed over approximately one month (PND28 to PND60).

### 4.3. Experimental Design

#### 4.3.1. Evaluation of Emotional/Cognitive Disturbances and Associated Neuromolecular Changes Induced by Perinatal Dronabinol Exposure

The first set of offspring (VEH: n = 49 females and 53 males; THC: n = 50 females and 55 males) was employed to evaluate the consequences of the exposure to dronabinol during pregnancy and lactation on (1) emotional behavior, cognitive states, and pre-attention, (2) the HPA axis, reward circuit, and endocannabinoid system target gene expression (RT-PCR), and (3) NeuN, BDNF, NF200, and VGluT1/VGAT protein expression (immunohistochemistry).

#### 4.3.2. Evaluation of Oral Ethanol Self-Administration and Gene Expression in Animals Exposed to Dronabinol during the Perinatal Period

A second set of animals (VEH: n = 13 females and 12 males; THC: n = 14 females and 17 males) was employed to evaluate the consequences of the exposure to dronabinol during pregnancy and lactation for (1) oral ethanol self-administration and (2) reward circuit and endocannabinoid system target gene expression (RT-PCR).

### 4.4. Emotional Evaluation

#### 4.4.1. Light–Dark Box (LDB)

This test assesses rodent anxiety-like behaviors by exposing them to an aversive environment [82,83,84]. The LDB test was carried out using an apparatus with two methacrylate compartments (20 cm × 20 cm × 15 cm), one transparent and the other dark and opaque, separated by an opaque tunnel. The light compartment was illuminated with a lamp (60 W) placed 25 cm above it. The mice were placed in the lightbox facing the tunnel, and during 5 min sessions, the time spent in the lightbox and the number of transitions between both boxes were recorded. 

#### 4.4.2. Novelty-Suppressed Feeding Test (NSFT)

This test measures anxiety-induced hyponeophagia as inhibiting food intake in an anxiety-provoking environment [83,85]. After 24 h of food deprivation, the mice were placed in a transparent square cage (40 cm × 40 cm × 50 cm) with a single pellet in its center. The latency time before eating was recorded up to a threshold of 5 min. Once the mouse began eating, the food pellet consumption was measured for another 5 min. 

#### 4.4.3. Tail Suspension Test (TST)

This is a widely accepted test for evaluating rodent depressive-like behaviors [84,86]. The mice were suspended individually by the tail at the edge of a lever above the tabletop (at 35 cm) using adhesive tape approximately 1–2 cm from the tip of the tail [86]. The immobility time was measured for 6 min. In this situation, mice develop escape-oriented behaviors interspersed with temporally increasing periods of immobility. 

#### 4.4.4. Prepulse Inhibition (PPI)

The acoustic prepulse inhibition paradigm was used to evaluate pre-attentional deficits in male and female mice exposed to dronabinol during gestation and lactation following the protocol described previously [87]. Soundproof chambers equipped with loudspeakers and controlled by the startle and fear conditioning System, PACKWIN 2.0 Software (Panlab, Barcelona, Spain) were employed. The animals were placed inside a plexiglass cylinder, and a piezoelectric accelerometer was used to transform movements into digital signals. For acclimatization, 3 days before the test sessions, the mice were handled and placed in the apparatus for 5 min per day without background noise. Test sessions began with a habituation phase by placing a subject undisturbed for 10 min with constant 65 dB background noise. After, each subject was presented with 80 trials over the 37 min test interval, including eight trial types in pseudorandom order: 1 × 40 ms, 120 dB acoustic startle stimulus, 3 × 20 ms prepulse stimulus (74, 82 and 90 dB), and 3 × 20 ms prepulse (100 ms before the onset) plus startle stimulus trials. Finally, trials with no stimulus were used to measure baseline movements. The average inter-trial interval was 15 s, and the maximum startle amplitude was recorded during the 100 ms sampling window. The mean percentage of prepulse inhibition achieved with each prepulse intensity and the mean startle amplitude during pulse-alone trials were analyzed. The ratio of the startle response was calculated as the ratio between the (startle response on acoustic prepulse + startle stimulus) trials and the startle-response-alone trials. The prepulse inhibition percentage was 100 × (1 − the ratio of startle response).

### 4.5. Cognitive Evaluation

#### 4.5.1. Novel Object Recognition (NOR)

This is an accepted and frequently used test for assessing recognition memory [88]. During the habituation phase, the mice were exposed to a square cage with two identical objects (named familiar objects), and the exploration time of each object was recorded for 5 min. Similarly, the exploration time of both objects was measured for 5 min, calculating the discrimination index as Discrimination Index = (New object exploration time − Familiar Object exploration time)/(New object exploration time + Familiar Object exploration time). After 24 h, the mice were re-exposed to the same cage, switching one of the familiar objects for a new one.

#### 4.5.2. Step-Down Inhibitory Avoidance (SDIA)

This test was described by Izquierdo and colleagues [89]. It consisted of an acrylic box (50 cm × 25 cm × 25 cm) with a floor composed of a grid of parallel stainless-steel bars, 1 mm in diameter and 1 cm apart. In the habituation phase, the animals were placed on the platform, and immediately upon stepping down, they received a foot shock (0.5 mA for 2 s). The mice were returned to the home cage after 24 h to evaluate their long-term aversive memory.

### 4.6. Oral Ethanol Self-Administration (OEA)

The OEA paradigm was carried out in 22 operant chambers (Panlab, Barcelona, Spain) [90] equipped with a chamber light, two levers, one receptacle for dropping the liquid solution, one syringe pump, one stimulus light, and one buzzer. PACKWIN 2.0 software (Panlab, Barcelona, Spain) controlled the stimulus and fluid delivery and recorded operant responses (number of presses of chamber levers). When the mice pressed one lever, it did not have any consequences (inactive lever), whereas pressing the other lever delivered 36 µL of fluid combined with a 0.5 s stimulus light and a 0.5 s 2850-Hz 85 dB buzzer beep (active lever), followed by a 6 s timeout period. 

The experiment was divided into three phases: (1) Training, (2) Saccharine substitution, and (3) 8% (*v*/*v*) Ethanol consumption and motivation. After the saccharine substitution period, the animals underwent the following three stages to evaluate basal ethanol (8% *v*/*v*) consumption and motivation: (i) The fixed ratio 1 (FR1) reinforcement schedule: five daily 1 h sessions in which mice responding one time on the active lever released one reinforcement (36 µL of ethanol 8% (*v*/*v*)); (ii) The fixed ratio 3 (FR3) reinforcement schedule (five daily 1 h sessions in which mice responding three times on the active lever releases one reinforcement); and (iii) The progressive ratio (PR) reinforcement schedule (one 2 h session in which the mice responding requirement to gain reinforcements increased according to the following series: 1-2-3-5-12-18-27-40-60-90-135-200-300-450-675-1000. During the PR session, the breaking point was obtained, corresponding to the maximum number of activations of the active lever that each animal could perform to achieve one reinforcement. 

### 4.7. Conventional and Confocal Immunohistochemistry

This study focused on HIPP’s dentate gyrus (DG), cornu ammonis 3 (CA_3_), and cornu ammonis 1 (CA_1_) region, and somatosensorial cortex. The general criteria reported by Amaral and Witter were used to define the hippocampal areas and strata [61]. 

Briefly, 24 h after the last behavioral test evaluation, the adult male and female mice were weighed, anesthetized with isoflurane, and intracardially perfused with 50 mL of saline followed by 250 mL of 4% paraformaldehyde, 0.1 M sucrose, and 0.002% CaCl_2_ in 0.1 M phosphate buffer (PB; 1.4% K_2_HPO_4_ 14 g/L, NaH_2_PO_4_.2H_2_O ~ 3 g/L to pH 7.3–7.4). The brains were dissected, post-fixed by immersion in the same perfusion medium at room temperature for 4 h and then stored in 0.05% sodium azide in PB at 4 °C. Eight parallel series of coronal sections containing the rostromedial portion of the DG, the CA, and the parietal cortex (−1.7 to −2.18 from Bregma) were cut with a Microm HM 650 V vibratome (Thermo Fisher Scientific, Inc., Barcelona, Spain) at 50 μm and stored in 0.05% sodium azide in PB at 4 °C. One series was immunostained with anti-mature neurons neuronal nuclei (NeuN) monoclonal antibody (mAb) (1:400; Merck KGaA, Darmstadt, Germany). Immunolabeled sections were incubated with biotinylated horse anti-mouse Ab (1:150), a Vectastain ABC kit (1:200; both from Vector Laboratories, Inc., Burlingame, CA, USA), and 0.05% 3,3′diaminobenzidine (DAB, Sigma-Aldrich Co., St. Louis, MI, USA). The sections were mounted on gelatinized slides and air-dried for 24 h, dehydrated in ethanol, cleared in xylol, and coverslipped. The adjacent series was double immunostained for fluorescence, starting with guinea pig anti-VGluT1 antibody (1:5000; Synaptic Systems; North Saanich, BC, Canada) and then rabbit anti-VGAT antibody (1:2000; Synaptic Systems; North Saanich, BC, Canada). All sections were incubated with goat anti-guinea pig antibody and Alexa Fluor 488-labeled (1:200, Molecular Probes, Invitrogen, Barcelona, Spain), followed by goat biotinylated anti-rabbit antibody Cross-Adsorbed Secondary Antibody, Alexa Fluor™ 405 (1:200, Invitrogen). 

Rabbit anti-neurofilament 200 antibody (NF200; 1:500; Sigma-Aldrich) was used to label the neurofilament and rabbit anti-brain delivery neurotrophic factor antibody (BDNF) (1:400, Millipore Corp, Darmstadt, Germany). NF200-ir and BDNF-ir sections were then incubated with goat anti-rabbit antibody, Alexa Fluor 488 (1:200, Molecular Probes, Invitrogen, Barcelona, Spain).

The sections were mounted using ProLong Gold (Molecular Probes, Invitrogen). Six coronal sections per group and sex were analyzed for each immunohistochemistry. All samples were examined in a Leica SPEII confocal laser fluorescence microscope, with images captured using Leica Application Suite X (LAS X) software.

### 4.8. Gene Expression Analyses by RT-PCR

The relative gene expression was analyzed in two different batches of animals killed by cervical dislocation: (set 1) at the end of the experimental procedures to evaluate emotional and cognitive behavior (PND60) and (set 2) after carrying out the OEA (PND100). In animals from set 1, the corticotropin-releasing factor (*Crf*) in the paraventricular nucleus (PVN), glucocorticoid receptor (*Nr3c1*) in the HIPP, cannabinoid receptors 1 (*Cnr1*) and 2 (*Cnr2*) in the HIPP and amygdala (AMY), mu-opioid receptor (*Oprm1*) in the nucleus accumbens (Nacc), and tyrosine hydroxylase (Th) in the ventral tegmental area (VTA) were analyzed by real-time polymerase chain reaction (RT-PCR). In animals from set 2, *Cnr1*, *Cnr2*, and *Oprm1* in the NAcc and Th in the VTA were analyzed by RT-PCR. Brains were removed from the skull and frozen at −80 °C. These samples were cut in a cryostat (−10 °C), obtaining coronal sections of 500 µm following the atlas of Paxinos and Fraklin [83,91]. They were microdissected following the procedure described by Palkovits and previously performed by our group [92,93]. Total ribonucleic acid (RNA) was extracted with TRI Reagent extraction, and reverse transcription was carried out (Applied Biosystems, Madrid, Spain). Quantitative analyses of the relative gene expression of *Crf* (Mm01293920_s1), *Nr3c1* (Mm00433832_m1), *Cnr1* (Mm00432621_s1), *Cnr2* (Mm00438286_m1), *Oprm1* (Mm01188089_m1), and *Th* (Mm00447546_m1) genes were performed on the StepOne Sequence Detector System (Applied Biosystems, Madrid, Spain). Data for each target gene were normalized to the endogenous reference gene 18S (Mm03928990_g1), and the fold change in the target gene expression was determined using the 2^−ΔΔCt^ method [94].

### 4.9. Statistical Analyses

Statistical analyses were performed using (i) Student’s *t*-test for comparing males and females, as well as animals exposed to VEH or THC, in the behavioral paradigms aimed to evaluate emotional (EPM, NSFT, TST) and cognitive (NOR) traits, in the PR stage (OEA), and the gene expression studies (RT-PCR); (ii) One-way analysis of variance (ANOVA) for comparing VEH- and THC-treated groups in the SDIA and PPI procedures; and (iii) two-way ANOVA with repeated measures (RM) followed by the Student–Newman–Keuls post hoc test for comparing VEH- and THC-treated male and female mice at different time points during the FR1 and FR3 stages (OEA). Differences were considered significant if the error probability was less than 5%. SigmaPlot 11 software (Systat Software Inc., Chicago, IL, USA) was used for all statistical analyses. 

## Figures and Tables

**Figure 1 ijms-25-07453-f001:**
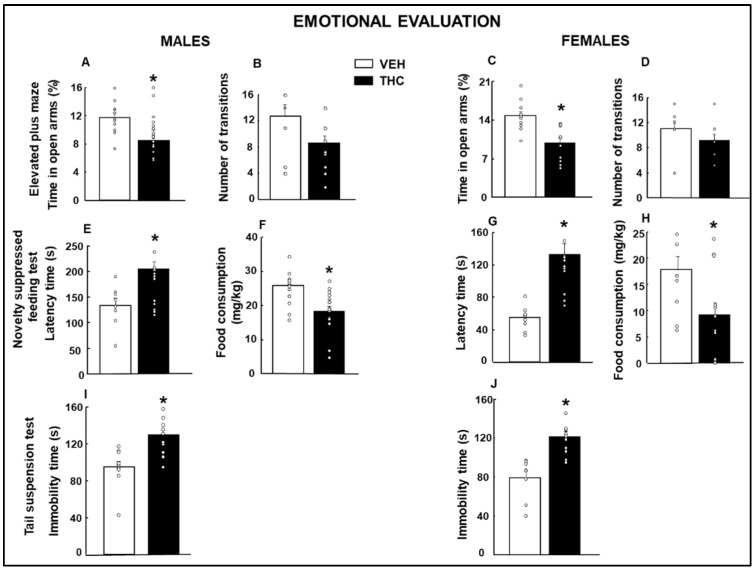
Emotional changes in male and female mice exposed to dronabinol (THC) or its vehicle (VEH) during gestation and lactation. The emotional state was measured from PND 28 and the following weeks until PND60. The elevated plus maze test was used to analyze anxiety-like behaviors by the (time in the open arms: (**A**,**C**), number of transitions: (**B**,**D**), and the novelty-suppressed feeding test (latency time: (**E**,**G**)) tests. Depressive-like behaviors were evaluated by the novelty-suppressed feeding (food consumption: (**F**,**H**)) and tail suspension (immobility time: (**I**,**J**)) tests. * *p* < 0.05, THC- vs. VEH-treated group.

**Figure 2 ijms-25-07453-f002:**
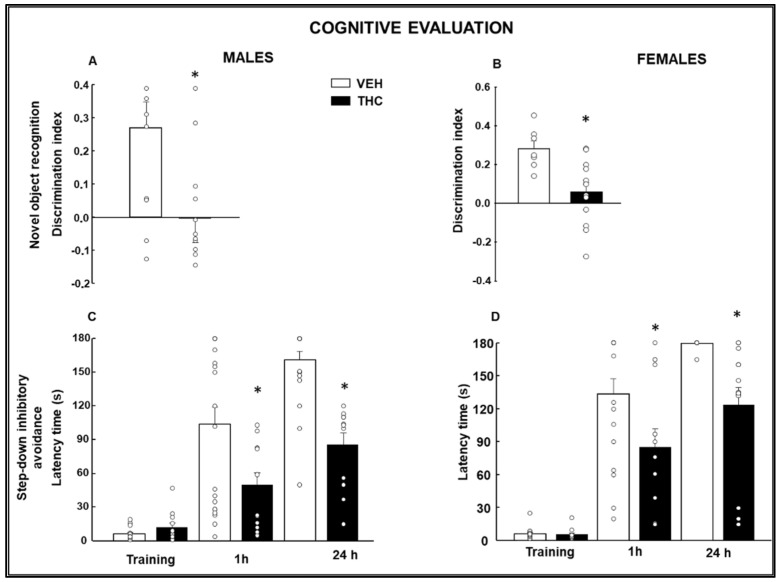
Cognitive changes in male and female mice exposed to dronabinol (THC) or its vehicle (VEH) were measured during gestation and lactation. The cognitive performance was evaluated from PND 28 and the following weeks until PND60. The novel object recognition (NOR; discrimination index: (**A**,**B**)) and the step-down inhibitory avoidance (SDIA; latency time to descend the platform at training 1 h and 24 h: (**C**,**D**)) tests were used to measure recognition and emotional memories, respectively. * *p* < 0.05, THC- vs. VEH-treated pups.

**Figure 3 ijms-25-07453-f003:**
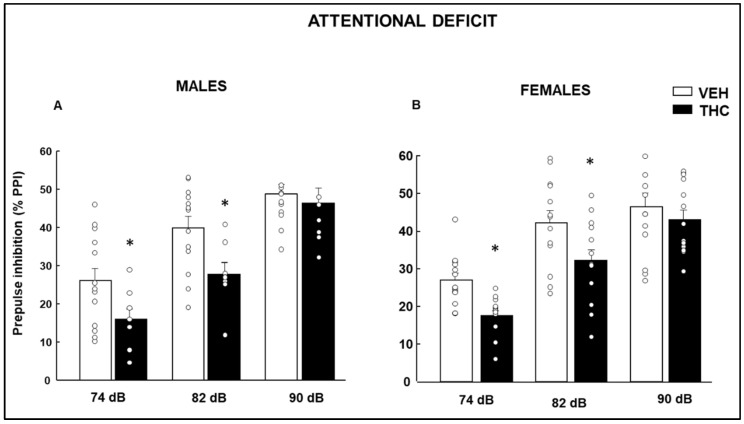
Evaluation of pre-attentional deficits by the % of prepulse inhibition of the acoustic startle response paradigm using 74, 82, and 90 dB prepulse stimuli in male (**A**) and female (**B**) mice exposed to dronabinol (THC) on gestation and lactation or its vehicle (VEH). * *p* < 0.05, THC- vs. VEH-treated group.

**Figure 4 ijms-25-07453-f004:**
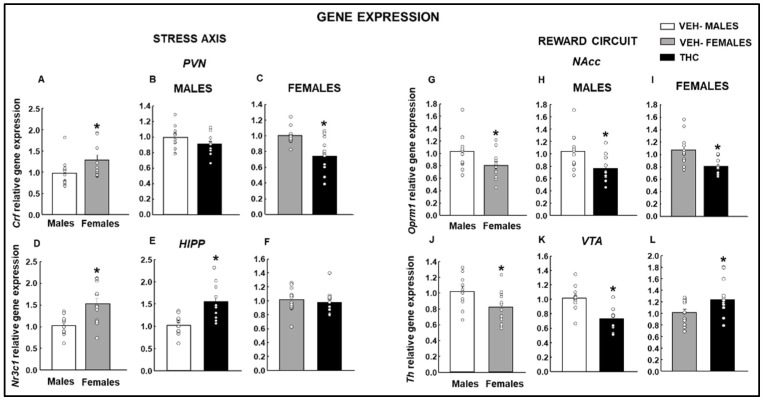
Analysis of relative gene expression by qPCR in the offspring of males and females exposed to dronabinol (THC) or VEH. Gene expressions of corticotrophin-releasing factor (*Crf*; **A**–**C**) in the paraventricular nucleus (PVN), glucocorticoid receptor (*Nr3c1*; **D**–**F**) in the HIPP, mu-opioid receptor 1 (*Oprm1*; **G**–**I**) in the nucleus accumbens (NAcc) and 2 (Cnr2; **J**–**L**), and tyrosine hydroxylase (Th; **J**–**L**) in the ventral tegmental area (VTA) were evaluated. * *p* < 0.05, VEH-treated female vs. male or THC- vs. VEH-treated groups.

**Figure 5 ijms-25-07453-f005:**
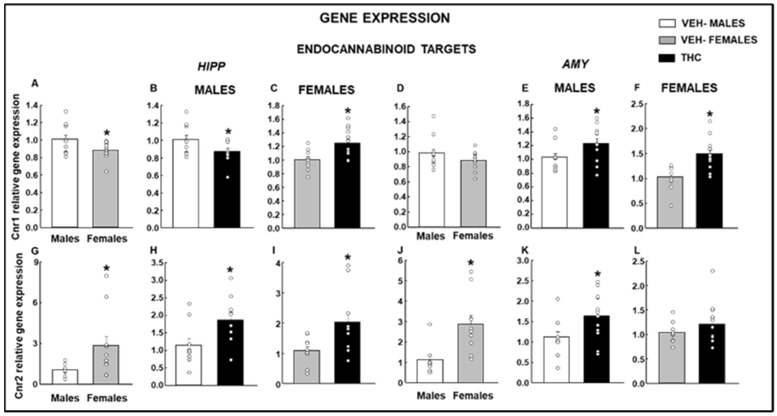
Analysis of relative gene expression by qPCR in the offspring of males and females exposed to dronabinol (THC) or VEH. Gene expressions of cannabinoid 1 receptor (*Cnr1*) in the HIPP (**A**–**C**) and amygdala (AMY; **D**–**F**) and of cannabinoid 2 receptor (*Cnr2*) in the HIPP (**G**–**I**) and AMY (**J**–**L**) were evaluated. * *p* < 0.05, VEH-treated female vs. male or THC- vs. VEH-treated groups.

**Figure 6 ijms-25-07453-f006:**
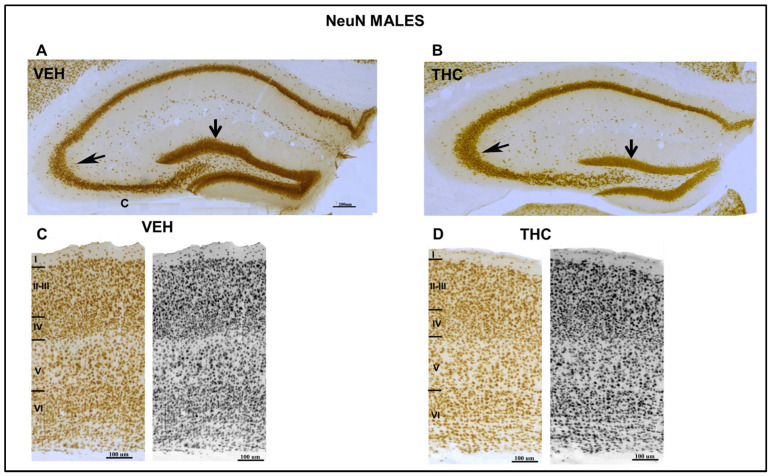
Low microscope magnification photomicrographs of NeuN-immunostained coronal sections of the HIPP and somatosensorial cortex in male mice. Image of NeuN-ir neurons in the VEH and the THC of male pups (**A**,**B**) at PND 60. A decrease in CA_3_ and DG (arrows) layers was observed in THC male (**B**) mice. Laminar organization of the somatosensory cortex of the VEH (**C**) and THC (**D**) offspring showed aberrant laminar organization of the cortex of THC pups compared to that of VEH mice. Note the increase in density in layer VI and the lower density in layers II–III in THC pups according to the migration pattern of the neocortex compared with that of the VEH mice—same scale for (**A**,**B**) and (**C**,**D**).

**Figure 7 ijms-25-07453-f007:**
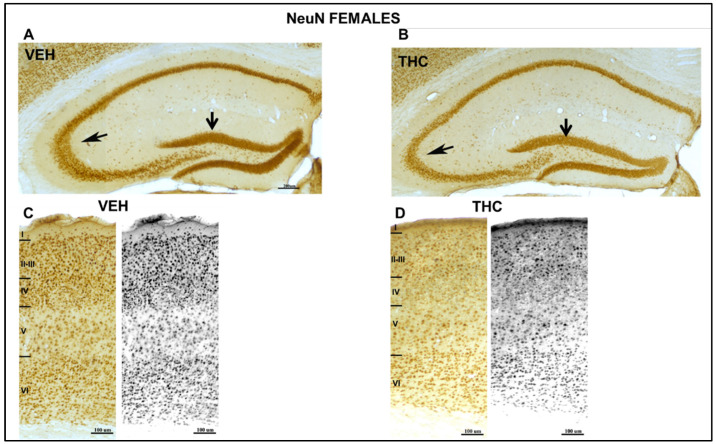
Low microscope magnification photomicrographs of NeuN-immunostained coronal sections of the HIPP and somatosensorial cortex in female mice. Image of NeuN-ir neurons in VEH and THC in female pups (**A**,**B**) at PND60 A decrease in CA_3_ and DG (arrows) layers was observed in THC (**B**) female mice. Laminar organization of the somatosensorial cortex of the VEH (**C**) and THC (**D**) offspring showed aberrant laminar organization of the cortex of THC pups compared with that of VEH mice. Note the increase in density in layer VI and the lower density in layers II–III in THC pups according to the migration pattern of the neocortex, compared with that of the VEH mice —the same scale for (**A**,**B**) and (**C**,**D**).

**Figure 8 ijms-25-07453-f008:**
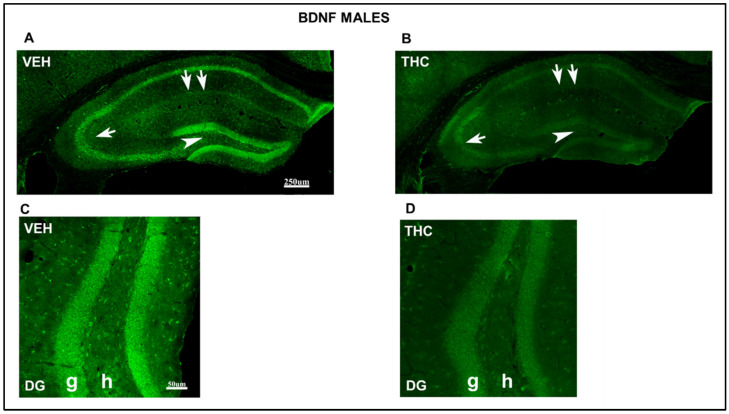
Confocal images indicating altered immunolabeling of BDNF in THC male pups. Confocal photomicrograph collages showing BDNF-ir (green labeling; **A**–**D**) in the HIPP of VEH (**A**,**C**) and THC (**B**,**D**) pups at PND60. Details of BDNF-in cells in the DG (**C**,**D**). Note the decrease in the number and immunostaining of BDNF-ir cells in the hilus and granular layer of DG (arrowhead in **A**,**B**), the pyramidal layer of CA1 (double arrow in **A**,**B**), and CA3 (arrow in **A**,**B**) in the lacunosum-moleculare layer in THC male pups compared to that in VEH mice. The results show a decrease in plasticity and cell survival in THC compared to VEH male pups. DG: dentate gyrus; h: hilus; g: granular layer—same scale for (**A**,**B**) and (**C**,**D**).

**Figure 9 ijms-25-07453-f009:**
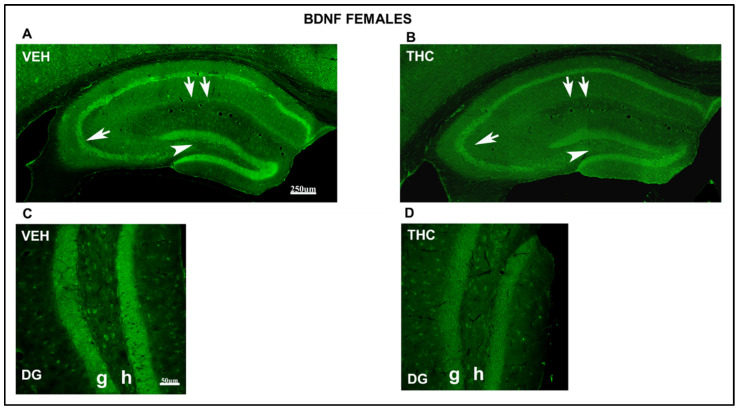
Confocal images indicating altered immunolabeling of BDNF in THC female pups. Confocal photomicrograph collages of exposing BDNF-ir (green labeling; **A**–**D**) in the HIPP of VEH (**A**,**C**) and THC (**B**,**D**) pups at PND60. Details of BDNF-in cells in DG (**C**,**D**). Note the decrease in the number of immunostained BDNF-ir cells in the pyramidal layer of CA_1_ (double arrow in **A**,**B**) and CA_3_ (arrows in **A**,**B**), in the hilus and granular layer of DG (arrowhead in **A**–**D**), and in the lacunosum-moleculare layer in THC female pups compared to those in VEH mice. The results show a decrease in plasticity and cell survival in THC compared with VEH female pups. DG: dentate gyrus; h: hilus; g: granular layer. The same scale applies to (**A**,**B**) and (**C**,**D**).

**Figure 10 ijms-25-07453-f010:**
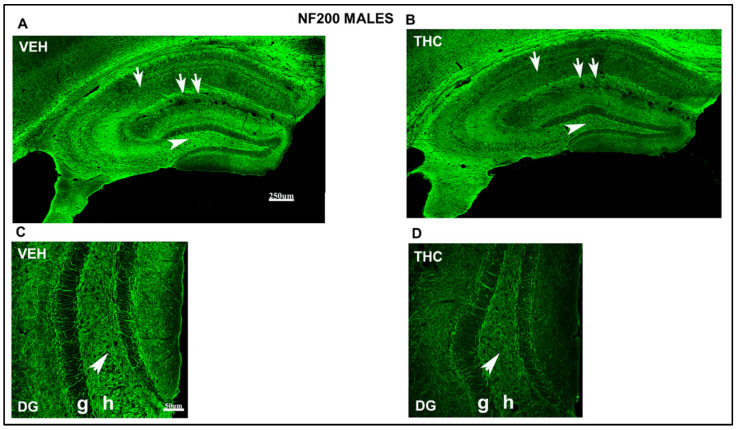
Confocal images at low magnification show abnormal immunolabeling of NF200 in THC male pups. NF200-ir (green labeling; **A**–**D**) in the HIPP of VEH (**A**–**C**) and THC (**B**,**D**) male mice at PND60. Note the decreased NF200-ir in the hillus of DG (arrowhead in **B**), the CA_1_ strata radiatum (arrow in **B**), and the stratum lacunosum-moleculare (double-arrow) in THC mice compared to that in VEH (**A**) male mice. Magnification of the hilus of the DG, the NF200-ir, is lower in the male mice of the THC group (**D**) than in the VEH group (**C**). Compared to VEH (**C**), these data indicate an alteration in the stabilization and maturation of connections in THC (**D**) male pups. DG: dentate gyrus; h: hilus; g: granular layer. The same scale applies to (**A**,**B**) and (**C**,**D**).

**Figure 11 ijms-25-07453-f011:**
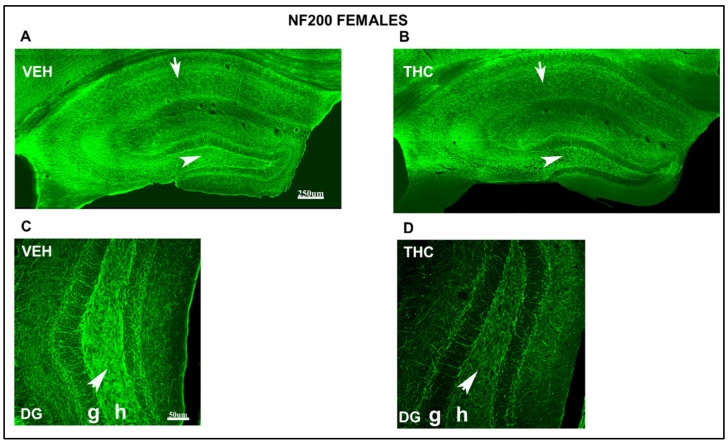
Confocal images at low magnification show abnormal immunolabeling of NF200 in THC female pups. NF200-ir (green labeling; **A**–**D**) in the HIPP of VEH (**A**–**C**) and THC (**B**, **D**) pups at PND60. Note the decreased NF200-ir in the CA1 strata radiatum (arrow in **B**) and the hilus of DG in THC mice (arrowhead in **B**) compared to that in VEH (**A**,**C**) females. The decrease in NF200-ir in lacunosum-moleculare is not as significant as in male THC mice (**B**). Magnification of the hilus of the DG, the NF200-ir, is lower in the THC group (**D**) than in the VEH group (**C**). Compared to VEH (**C**), these data indicate an alteration in the stabilization and maturation of connections in THC (**D**) female pups. DG: dentate gyrus; h: hilus; g: granular layer. The same scale applies to (**A**,**B**) and (**C**,**D**).

**Figure 12 ijms-25-07453-f012:**
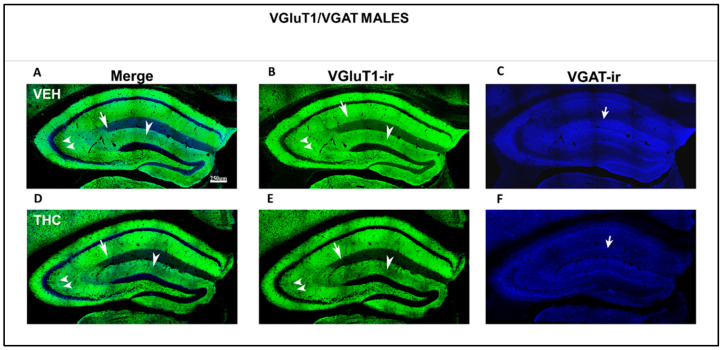
Low-magnification confocal images showing irregular VGluT1-ir and VGAT-ir in THC male pups. Confocal photomicrographs collage of VGluT1-ir (green labeling; **B**,**E**), VGAT-ir (blue labeling; **C**,**F**), and merged images (**A**,**D**) in the HIPP of VEH (**A**–**C**) and THC (**D**–**F**) male pups at P60. Observe the decreased VGluT1-ir in the DG distal-inner molecular layer (arrowhead in **D**,**E**), in the CA_3_ stratum lucidum (double arrow in **D**,**E**) and radiatum, and in the CA_1_ stratum lacunosum-moleculare (arrows in **D**,**E**) in THC compared to that in VEH male mice. These data show that the primary trisynaptic loop is irregular in THC male pups. A band of decreased VGAT-ir boutons at the lacunosum molecular layers of CA1 in THC pups is shown (arrow in **C**,**F**)—same scale for (**A**–**F**).

**Figure 13 ijms-25-07453-f013:**
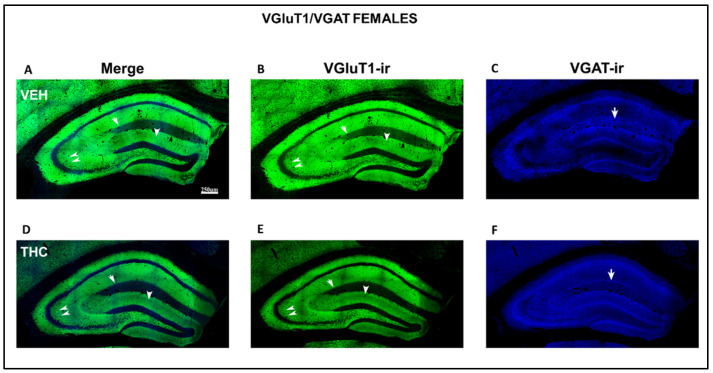
Low-magnification confocal images showing aberrant VGluT1-ir and VGAT-ir in female pups. Confocal photomicrograph collages showing VGluT1-ir (green labeling; **B**,**E**), VGAT-ir (blue labeling; **C**,**F**), and merged images (**A**,**D**) in the HIPP of VEH (**A**–**C**) and THC (**D**–**F**) female pups at P60. Observe the decreased VGluT1-ir in the DG distal-inner molecular layer (arrowhead in **D**,**E**), in the CA_3_ strata lucidum (double arrow in **D**,**E**) and radiatum, and in the CA_1_ stratum lacunosum-moleculare (arrows in **D**,**E**) in THC mice compared to that in VEH pups. These data show that the primary trisynaptic loop in THC female pups is irregular. A band of decreased VGAT-ir boutons at the lacunose-molecular layers of CA_1_ in THC pups is indicated (arrow in **C**,**F**)—the same scale for (**A**–**F**).

**Figure 14 ijms-25-07453-f014:**
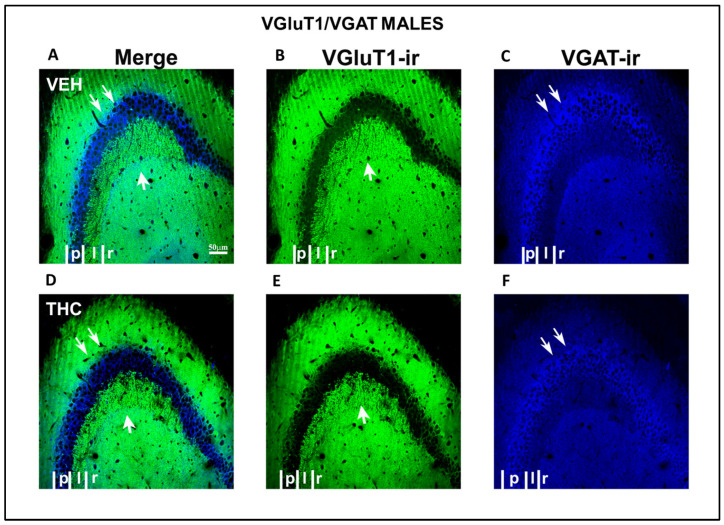
Confocal VGluT1-ir and VAGT-ir in CA_3_ of VEH and THC male pups. Images showing VGluT1-ir (green labeling; **B**–**E**), VGAT-ir (blue labeling; **C**–**F**), and merged images (**A**–**D**) in CA_3_ of VEH (**A**–**C**) and THC (**D**–**F**) male mice at P60. Observe that the area of VGluT1-ir mossy boutons in the stratum lucidum (**C**; arrow) of CA_3_ and VGAT-ir in the pyramidal layer (double arrow in **C**,**F**) is smaller in THC (**F**) mice than in VEH pups (**C**)—same scale for (**A**–**F**).

**Figure 15 ijms-25-07453-f015:**
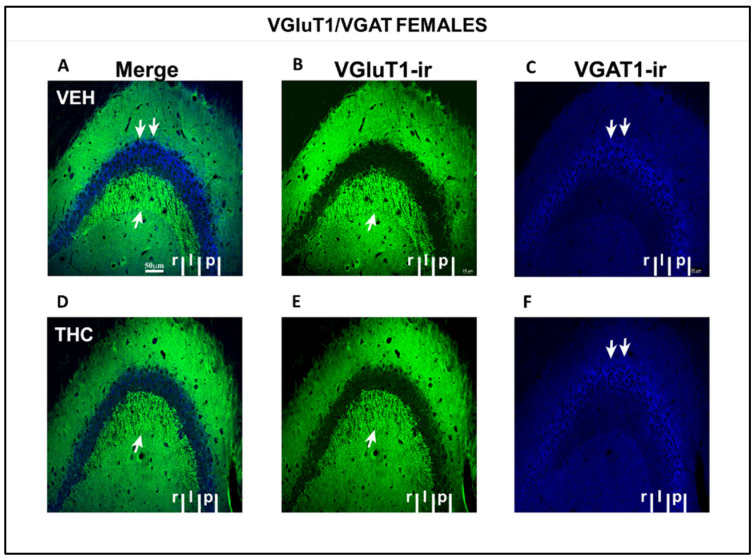
Confocal VGluT1-ir and VAGT-ir in CA_3_ of VEH and THC female pups. Images showing VGluT1-ir (green labeling; **B**–**E**), VGAT-ir (blue labeling; **C**–**F**), and merged images (**A**–**D**) in the CA_3_ of VEH (**A**–**C**) and THC (**D**–**F**) female pups at P60. Note that the area of VGluT1-ir mossy boutons in the strata lucidum (**C** arrow) of CA_3_ and VGAT-ir in the pyramidal layer (double arrow in **C**,**F**) is smaller in THC mice (**F**) than in VEH (**C**) pups—same scale for (**A**–**F**).

**Figure 16 ijms-25-07453-f016:**
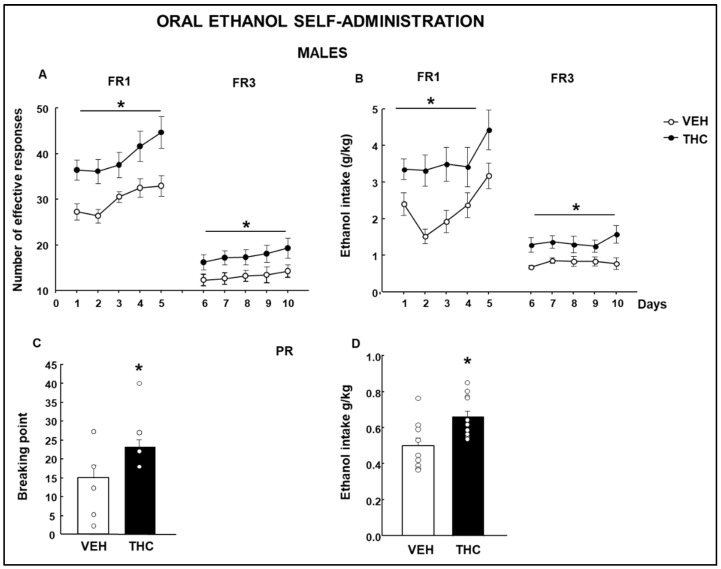
Oral ethanol self-administration evaluation in male mice perinatally exposed to dronabinol (THC) or its vehicle (VEH). Number of effective responses during the fixed ratio 1 (FR1) and fixed ratio 3 (FR3) stages (**A**). Ethanol intake is expressed as g/kg during the FR1 and FR3 stages (**B**). A breaking point was achieved, and ethanol intake was taken during the progressive ratio (**C** and **D**, respectively). * represents values from THC mice that are significantly different (**A**,**B**, two-way RM ANOVA, *p* < 0.05; **C**,**D**, Student’s *t*-test, *p* < 0.05) from those of the VEH-treated group.

**Figure 17 ijms-25-07453-f017:**
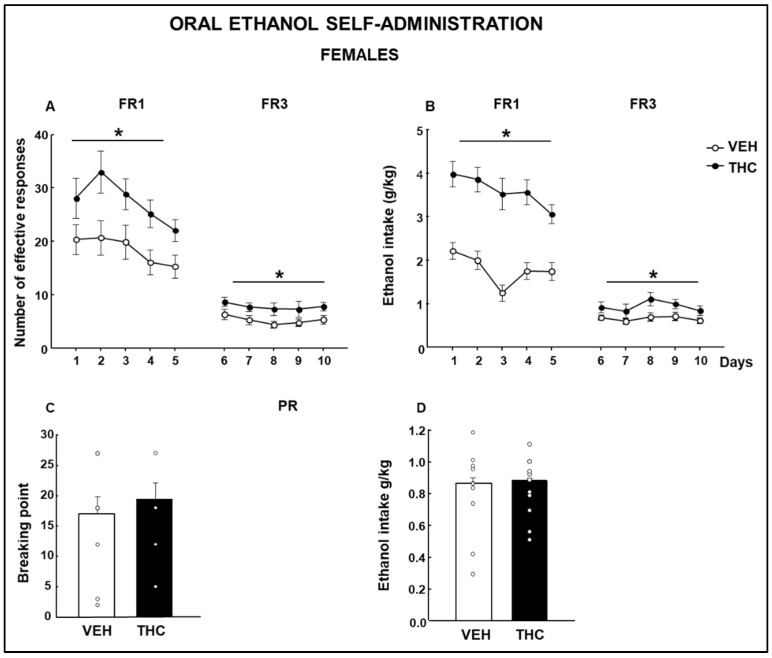
Oral ethanol self-administration evaluation in male mice perinatally exposed to dronabinol (THC) or its vehicle (VEH). Number of effective responses during the fixed ratio 1 (FR1) and fixed ratio 3 (FR3) stages (**A**). Ethanol intake is expressed as g/kg during the FR1 and FR3 stages (**B**). A breaking point was achieved, and ethanol intake was taken during the progressive ratio stage (**C** and **D**, respectively). * represents values from THC-treated mice that are significantly different (**A**,**B**, two-way RM ANOVA, *p* < 0.05) from those of the VEH-treated group.

**Figure 18 ijms-25-07453-f018:**
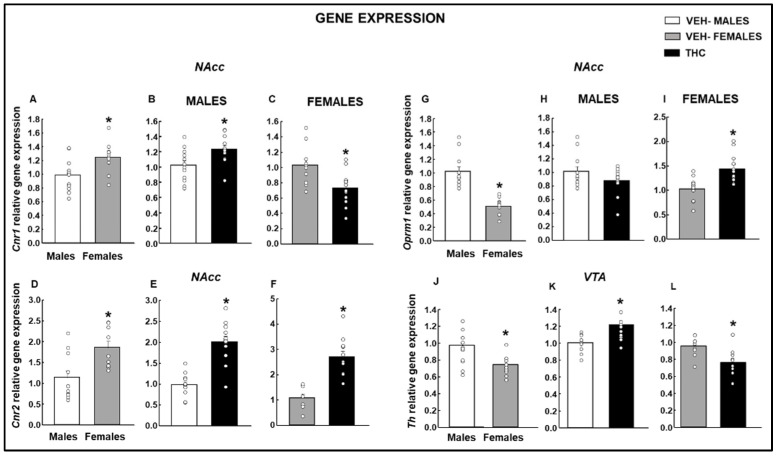
Analyses of relative gene expression by qPCR in male and female mice exposed to dronabinol (THC) or its vehicle (VEH) during gestation and lactation were evaluated in the OEA paradigm. Gene expression of *Cnr1* (**A**–**C**), Cnr2 (**D**–**F**), and *Oprm1* (**G**–**I**) in the NAcc, and Th (**J**–**L**) in the VTA were evaluated. * *p* < 0.05, VEH-treated female vs. male or THC- vs. VEH-treated groups.

**Figure 19 ijms-25-07453-f019:**
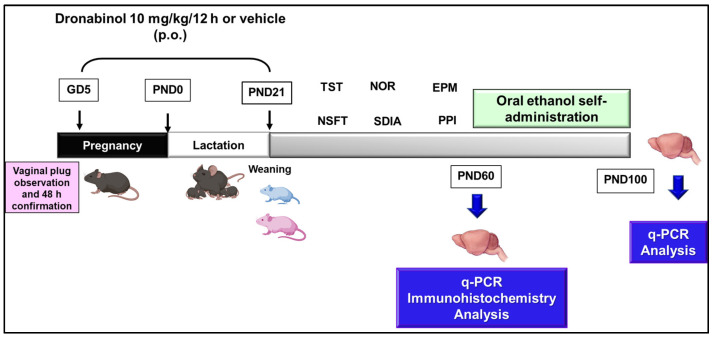
Summary of the schedule followed to develop the experimental procedures. After gestation confirmation, at GD5, dronabinol (or its vehicle) gavage was started at a dose of 10 mg/kg/12 h (p.o.) until the pup’s weaning at PND21. Behavioral evaluations were initiated one week after weaning to determine the presence of anxiety and depressive traits, cognitive alterations, and pre-attentional deficits using different experimental paradigms. At the end of these behavioral studies, mice were allocated to two different sets: (1) a first set was employed to evaluate the consequences of perinatal dronabinol exposure on the gene and protein expression of selected targets at PND60, and (2) a second set was employed to evaluate the consequences of perinatal dronabinol exposure on oral ethanol self-administration and the gene expression of selected targets at PND100. Mice from both sets were finally sacrificed, and frozen or perfused brain samples were obtained to perform the proposed molecular studies. GD: gestational day, PND: postnatal day, p.o.: per os (oral administration), TST: tail suspension test, NSFT: novelty suppressed feeding test, EPM: elevated plus maze, PPI: prepulse inhibition, NOR: novel object recognition, SDIA: step-down inhibitory avoidance.

## Data Availability

Data are contained within the article.

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
