# Peer review of "Fetal Cannabinoid Syndrome: Behavioral and Brain Alterations of the Offspring Exposed to Dronabinol during Gestation and Lactation"

_ijms, 2024, doi:10.3390/ijms25137453_

Round 1
Reviewer 1 Report
Comments and Suggestions for Authors
This study investigates the effects of dronabinol (THC) consumption during gestation and lactation on the behavioral and neurobiological development of offspring, focusing on sex-dependent differences. The study presents significant findings on the anxiogenic, depressive-like behaviors, cognitive disturbances, and alterations in gene expression and neurotransmission systems in exposed mice.
Limitations and Areas for Improvement
Although the study mentions sex-dependent differences, it lacks a detailed exploration of these patterns. The differential regulation of the HPA axis between sexes is discussed, but further clarification and detailed analysis of the mechanisms behind these differences would be beneficial.
The study evaluates changes up to postnatal day 60, but it does not address the potential long-term effects of perinatal dronabinol exposure on behavior and brain function. Discussion of longitudinal studies in the field would be valuable to assess the persistence of alterations reported in the study.
The findings in mice may not fully translate to humans. Paper should be strengthened by a paragraph discussing shortcomings of using the mouse model in this kind of studies.
In conclusion, the study provides significant insights into the adverse effects of dronabinol consumption during pregnancy and lactation on offspring, highlighting behavioral changes, cognitive impairments, and neurobiological alterations with a focus on sex-dependent differences. While the research is thorough and offers a comprehensive analysis, it could benefit from more detailed sex-specific data, exploration of long-term effects, and further mechanistic studies. Despite these limitations, the study underscores the critical need for caution regarding cannabis use during pregnancy and lactation, given the potential for severe and lasting impacts on offspring development.
Author Response
Dear reviewer,
Thank you for your comments, please find my reply in the attachment.

Reviewer 2 Report
Comments and Suggestions for Authors
Navarro et al. – IJMS:
The authors characterized a rodent model of high levels of perinatally THC exposure, taking into account for their analysis also sex differences in the offspring regarding behavioral and brain changes as well as the susceptibility to alcohol consumption in adulthood. The paper is well-written, and the authors chose the proper bibliography to support their data. Reason why, the paper is suitable for publication on IJMS.
Comments:
Results
Regarding the experiments for emotional evaluation and cognitive evaluation, in the figure legends, the authors wrote that evaluations were made “From PND 28 and during the following weeks until PND60”. Since the temporal window is long and includes pre-adolescence, adolescence, and early adulthood, I was wondering if the authors have done comparisons among these three different windows. If yes, did they find any difference? Moreover, considering the last phase, did the authors take into account the estrous cycle in female offspring, if yes, did they find any difference?
The authors stated that there is a decrease of neurons in CA3 and DG of THC male and female offspring, did they measure it for example as a % of decrease (if it is feasible)? Similarly, they assessed a higher density of cortical neurons in layer VI and a decrease in layers II-III of THC male and female offspring, did they check for a % of increase and decrease (again, only if it is feasible)?
Discussion
Could the authors explain better why they chose this high dose of THC, in terms of corresponding habits in humans (e.g. heavy smokers or mild smokers)?
To improve the take home message of the paper would be helpful if the authors discussed better why they chose defined brain areas and genes for their analysis.
Ln 513, pg 18, the authors mentioned the influence of circulating sexual hormones, did they find differences in females based upon their estrous cycle?
Ln 644, pg 20, the authors mentioned reduced synaptic plasticity, would be helpful if they explained with which experiments they evaluated it.
Methods
Ln 671-672, pg 21, the authors stated that only 60% of the females exposed perinatally to THC finished gestation and lactation periods properly. I was wondering if the authors checked also for litter size and male/female ratio to better characterize this exposure paradigm. Moreover, did the authors measure pup weight from birth to weaning in order to check for an effect in weight gain with this high dose of THC?
Author Response

(The authors gave the same response as above.)
